# Combining molecular dynamics simulations with small-angle X-ray and neutron scattering data to study multi-domain proteins in solution

Andreas Haahr Larsen[1,2¤a], Yong Wang[1], Sandro Bottaro[1¤b], Sergei Grudinin[3], Lise Arleth[2], Kresten Lindorff-Larsen[1]*

1 Structural Biology and NMR Laboratory, Linderstrøm-Lang Centre for Protein Science, Department of Biology, University of Copenhagen, Copenhagen, Denmark, 2 X-ray and Neutron Science, Niels Bohr Institute, University of Copenhagen, Copenhagen, Denmark, 3 Univ. Grenoble Alpes, CNRS, Inria, Grenoble INP, LJK, Grenoble, France

¤a Current address: Structural Bioinformatics and Computational Biochemistry Unit (SBCB), Department of Biochemistry, University of Oxford, Oxford, United Kingdom.
¤b Current address: Atomistic Simulations Laboratory, Istituto Italiano di Tecnologia, Genova, Italy.
* lindorff@bio.ku.dk

**Data Availability Statement:** All analysis scripts and data are available at https://github.com/KULL-

## Abstract

Many proteins contain multiple folded domains separated by flexible linkers, and the ability to describe the structure and conformational heterogeneity of such flexible systems pushes the limits of structural biology. Using the three-domain protein TIA-1 as an example, we here combine coarse-grained molecular dynamics simulations with previously measured small-angle scattering data to study the conformation of TIA-1 in solution. We show that while the coarse-grained potential (Martini) in itself leads to too compact conformations, increasing the strength of protein-water interactions results in ensembles that are in very good agreement with experiments. We show how these ensembles can be refined further using a Bayesian/Maximum Entropy approach, and examine the robustness to errors in the energy function. In particular we find that as long as the initial simulation is relatively good, reweighting against experiments is very robust. We also study the relative information in X-ray and neutron scattering experiments and find that refining against the SAXS experiments leads to improvement in the SANS data. Our results suggest a general strategy for studying the conformation of multi-domain proteins in solution that combines coarse-grained simulations with small-angle X-ray scattering data that are generally most easy to obtain. These results may in turn be used to design further small-angle neutron scattering experiments that exploit contrast variation through $^1$H/$^2$H isotope substitutions.

## Author summary

Many proteins contain multiple folded domains separated by flexible linkers, and in order to understand how such multi-domain proteins function, we need to be able to describe

Centre/papers/tree/master/2020/TIA1-SAS-Larsen-et-al.

**Funding:** The project was supported by the Lundbeck Foundation BRAINSTRUC initiative in structural biology (LA, KL-L), the Nordforsk Nordic Neutron Science Programme (LA, KL-L) and a Carlsberg foundation internationalization stipend (AHL). The funders had no role in study design, data collection and analysis, decision to publish, or preparation of the manuscript.

**Competing interests:** The authors have declared that no competing interests exist.

how these domains are oriented in space. We have used the three-domain protein TIA-1 as an example to combine molecular simulations with biophysical experiments to describe the structural and dynamical properties of a multi-domain protein. We show that while standard simulations do not lead to good agreement with the experimental data, we can improve the agreement substantially by tuning a single parameter in the model that describes the interaction between protein and water. We can gain further information about the system by a more direct integration of the data, and we find that we can provide a detailed and robust description of the relative location of the different domains in TIA-1. The method is general and will be useful to study the relationship between structure, dynamics and function in multi-domain proteins in other systems.

## Introduction

The ability to change conformation is crucial to the function and regulation of many proteins, and describing and quantifying protein flexibility is important when studying the function of proteins and their complexes. Examples of such dynamics includes flexibility through a hinge region, or the movement of domains connected by flexible linkers [1]. The extreme case is highly entropic systems such as intrinsically disordered proteins. Many experimental methods for studying protein structure are, however, only indirectly sensitive to structural flexibility, or may even suppress or bias dynamical properties. In X-ray crystallography, flexible regions in termini or loops are often removed before crystallization, as they may hinder precipitation and formation of protein crystals. Even when left in the construct, flexible parts may not be visible in the final refined model, resulting in models for the folded parts only. Although cryo-electron microscopy is in principle a single-molecule technique, it is in practice also difficult to define flexible parts, as these may average out when refining 2D and 3D models.

Solution NMR and small-angle X-ray scattering (SAXS) are two widely used techniques that can be used to study protein flexibility and dynamics in solution. Where NMR generally contains information about the relative orientation of atoms that are close in space (with residual dipolar couplings representing a notable exception), SAXS carries information on the overall protein structure. Therefore, SAXS is particularly useful when the structure of the individual domains of a multi-domain protein has been solved by high-resolution methods, but the structure of the full-length protein and the relative orientations of the domains remain unknown. While it may in certain cases be possible to fit the data with a single protein structure, the resulting structure may be a biased representation of a flexible protein with many different conformations with different occupancies.

One approach to generate such conformational ensembles is to use molecular dynamics (MD) simulations. Despite progress in both sampling methods and molecular force fields, such simulations may still give rise to conformational ensembles that are not in perfect agreement with experimental data. In that case, however, simulations and experiments may be used synergistically to generate and refine the description of flexible molecules. Thus, as described by us and others, SAXS and molecular simulations can be combined to determine a structural ensemble that represents the system, and is compatible with the information in the force field and the experimental constraints from SAXS [2–20].

Here we apply the Bayesian/Maximum Entropy (BME) method [3] to integrate simulations and small-angle scattering data from a flexible multi-domain protein. We used the coarse-grained force field Martini [21] for the MD simulations to overcome sampling issues, which is particular relevant for larger and conformationally heterogeneous systems [14]. We find that, despite recent improvements of Martini 3 [22], the Martini force-field needs to be adjusted to

provide a better fit to the SAXS data, and that this can be performed by changing the strength of protein-water interactions. Moreover, we show how the BME reweighting protocol can be used to obtain full consistency with data, both for the force field refined against data, and for force fields that give rise to greater discrepancies with the data.

We also investigate SAXS in combination with the related technique, small-angle neutron scattering (SANS). In particular we discuss how SANS can contribute with information to analyse the distribution of conformations of flexible proteins. Substitution of hydrogen with deuterium in the protein and/or solvent changes the excess scattering length density, or contrast, in SANS with each contrast carrying different structural information. Thus, SANS measurements are potentially interesting for multi-domain proteins, as they allow the investigator to highlight individual domains by contrast variation [23]. Specifically, we extend the work from Sonntag et al. [24] who used a combination of SAXS and contrast variation SANS data to refine individual conformations of a multi-domain protein. Building also upon recent work by Chen et al. [7], we here use the SAXS and SANS data to determine several ensembles of conformations. We analyse each of the SANS contrasts measured by Sonntag et al. [24], and examine what information is carried by them. We also discuss how a SANS experiment could potentially be further optimized regarding choice of contrast situations, such that the information gain can be maximized.

We have chosen the three-domain protein TIA-1 as a model system for our analyses. The three folded domains of TIA-1, RNA recognition motifs 1, 2 and 3 (RRM1, RRM2, and RRM3), are connected by linkers that provide a high degree of structural flexibility to the complex, and high-resolutions structures exist of all domains. Both SAXS and SANS data were measured and previously analysed by Sonntag et al. [24], who used segmental domain-wise perdeuteration of the domains in TIA-1 and mixtures of $H_2O$ and $D_2O$ in the solvent to obtain the different SANS contrasts. We find that while simulations with the Martini coarse-grained force field lead to imperfect agreement with experiments, strengthening the protein-water interactions in the Martini potential enables relatively accurate fitting of the data. An even better agreement can be obtained by using a Bayesian/Maximum Entropy approach to fit the experimental data, and we show that fitting ensembles to the SAXS data leads to an improved agreement with the SANS data.

## Methods

### Generating the initial structure for MD simulations

Native TIA-1 has three RNA recognition motifs, RRM1, RRM2, and RRM3, connected by flexible linkers and followed by a C-terminal unstructured Q-rich domain of ~100 residues. In this study, we investigated a truncated construct of TIA-1 without the Q-rich domain [24], and will in the following refer to this truncated construct simply as TIA-1. As starting point for our models, we used previously determined high-resolution structures of the three folded domains. The structure of RRM1 was determined by solution state NMR spectroscopy [24] (PDB 5O2V), the structure of RRM2 was determined by X-ray crystallography [24] (PDB: 5O3J), and the structure of the RRM2-RRM3 complex (RRM23) was determined by solution state NMR spectroscopy [25] (PDB: 2MJN). We added missing residues, in particular in the linker between RRM1 and RRM23, using Modeller [26] to generate the initial model for the MD simulations (Fig 1A).

### Setting up the MD simulations

We used the coarse-grained force field Martini version 3.0.beta.4.17 [21,22] in combination with GROMACS 5.1.4 or 2016.5 [27]. First, the structure was coarse-grained using the

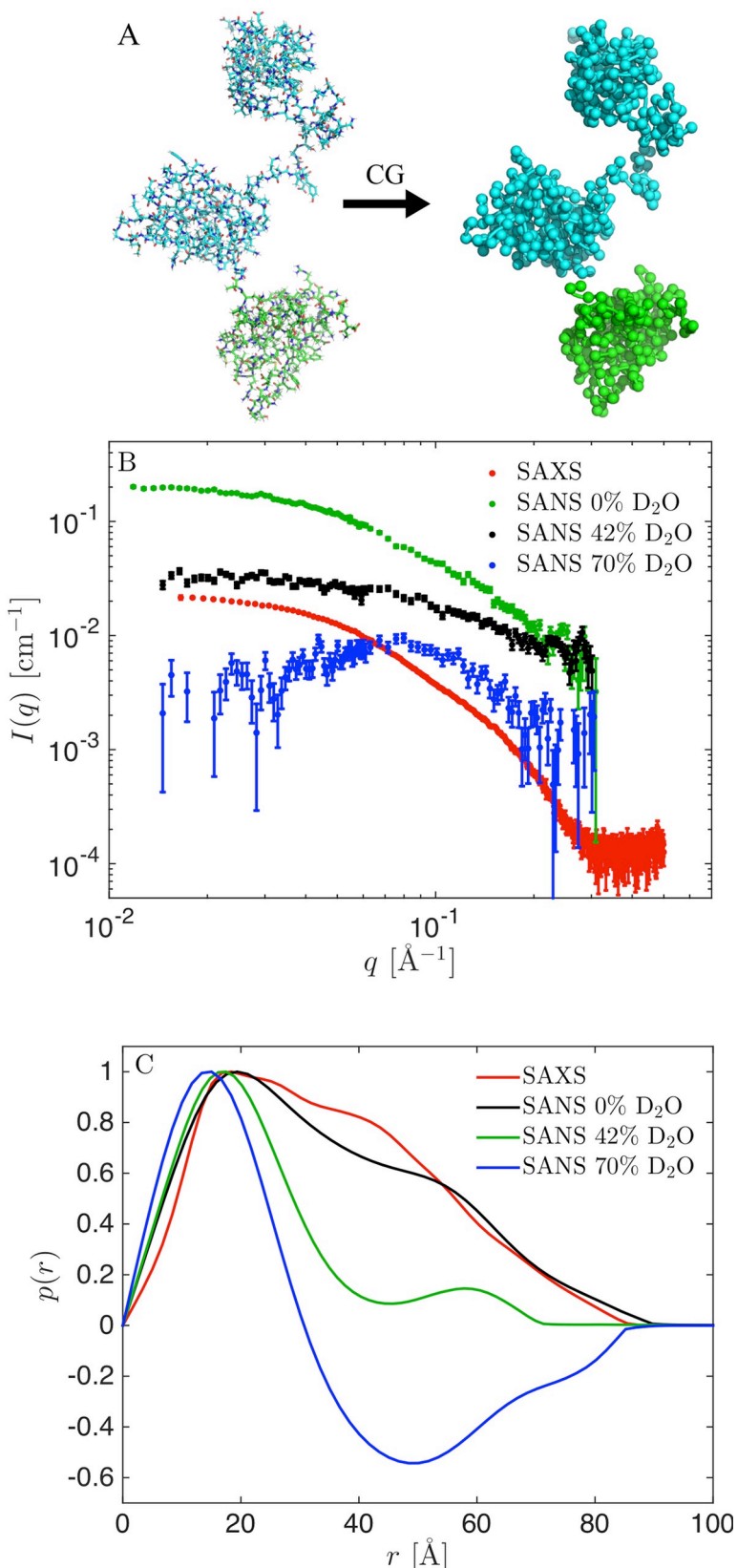

**Fig 1. TIA-1 structure and experimental SAXS and SANS data.** (A) All-atom model of TIA-1 and the corresponding coarse-grained Martini model. (B) Experimental SAXS (black) and SANS data (green: 0% $D_2O$, red: 42% $D_2O$, blue: 70% $D_2O$). (C) Corresponding pair distance distribution functions, $p(r)$.

Martinize2 python script [28] (Fig 1A). An elastic network [29] was then added to the folded domains, to make them semi-rigid. Specifically, a harmonic potential with a force constant of 500 kJ $mol^{-1}$ $nm^{-2}$ was applied to all backbone (BB) bead pairs with relative distance less than 0.9 nm. No elastic network was added to the flexible linkers or to contacts between the three folded domains, so that we only applied restraints within RRM1 (residues 7–80), RRM2 (residues 95–170) and RRM3 (residues 204–270). The coarse-grained structure with the elastic network was then relaxed for 3 ps, with a time step of 30 fs, using the Berendsen thermostat and barostat, and Verlet cutoff scheme. The relaxed structure was equilibrated for 1 ns, with a time step of 5 fs, using the velocity-rescale (v-rescale) thermostat [30], Parinnello-Rahman barostat, and Verlet cutoff scheme.

## Running the MD simulations

Production runs were initiated using the equilibrated structure and run for 10 μs with a time step of 20 fs, using v-rescale thermostat, Parinnello-Rahman barostat, and Verlet cutoff scheme in the NPT ensemble. Performance was 600–750 ns/day on four CPU cores. Frames were written every ns resulting in a total of 10,000 frames in each final trajectory.

## Calculating collective variables

We used PLUMED version 2.4.1 [31] to calculate the radius of gyration, $R_g$, along with the distances between the centres of mass of the three domains, $D_{12}$, $D_{13}$ and $D_{23}$ respectively. Error estimates were determined with block analysis [32].

## Backmapping from coarse-grained to all-atom

As described below, we calculated SAXS and SANS data using software that takes as input all-atom structures. Thus, we "back-mapped" from coarse-grained to all-atom using a modified version of a backmapping algorithm for Martini [33]. The original algorithm consists of two energy minimization runs with 500 steps followed by five simulation runs with increasing time steps from 0.2 fs to 2 fs. As our goal was simply to calculate SAXS and SANS data from these structures, we simplified the algorithm by leaving out the five simulation runs and limit the number of steps in the energy minimization runs to 200, and in this way reduced the computational cost. We tested that this simplification did not affect calculated SAXS curves substantially, as judged by comparison of calculated curves after the full back-mapping algorithm and after the simplified algorithm (S1 Fig). The simplified algorithm also did not have any substantial impact on the radius of gyration, $R_g$, as calculated from the SAXS data. Higher resolution differences, not immediately detectable by SAXS, could however be seen when comparing the back-mapped structures from the full and simplified back-mapping procedures. The simplification resulted in an ~80% reduction of the computation time for the back-mapping procedure. With the simplified algorithm, the calculation time for back-mapping 10,000 frames was about 50 hours using a node with 8 cores.

## Calculating SAS curves

SAXS and SANS intensity curves, $I(q)$, where the momentum transfer $q = 4\pi \sin(\theta)/\lambda$, is given via the wavelength $\lambda$ and scattering angle $2\theta$, were calculated using Pepsi-SAXS 2.4 and Pepsi-

SANS 2.4 [34,35]. Resolution effects were included in the Pepsi-SANS calculations using the uncertainty of the measured $q$-values. Specifically, we applied a Gaussian convolution to the theoretical $I(q)$ curve with a Gaussian of the form, $\exp(-q^2/2\sigma_q^2)$, where $\sigma_q$ is the standard deviation taken from the forth column of the experimental SANS data. In both SAXS and SANS, the forward scattering, $I(0)$, was fitted along with the excluded water volume, $r_0$, the density of the hydration shell, $\Delta\rho$, and a constant background, $B$. Fitting all four parameters freely and for each conformation could lead to a drastic overfit of the data. Therefore, for each ensemble (set of conformations and associated weights) we estimated a single set of global values for these parameters using the following algorithm:

1. Each frame was fitted individually, with the four parameters free.

2. Trajectories were reweighted using BME (see details below) using a range of $\theta$-values from 1 to 500. This resulted in a set of weights for each value of $\theta$: $\{w_\theta\} = \{w_{\theta,1}, w_{\theta,2}, \ldots, w_{\theta,N})$, where $N$ is the number of frames.

3. Weighted averaged parameter values were calculated using:

$$\langle p \rangle_w = \sum_i w_{\theta,i} \cdot p_i,$$

where $p$ is either $I(0)$, $B$, $r_0$, or $\Delta\rho$, and $i$ runs over all frames.

4. The scattering was calculated again for each frame using Pepsi-SAXS/SANS, with the parameters fixed to the weighted average.
   This resulted in a reduced $\chi^2$ for each $\theta$:

$$\chi_{r,\theta}^2 = \frac{1}{M-2} \sum_j \left( \frac{S \cdot \langle I_\theta(q_j) \rangle + B - I_{\text{exp}}(q_j)}{\sigma_j} \right)^2,$$

where $M$ is the number of data points, and $I_\theta(q_j)$ is the weighted average of the intensities:

$$\langle I_\theta(q_j) \rangle = \sum_i w_{\theta,i} \cdot I_{calc,i}(q_j).$$

The scale parameter, $S$, and the constant background, $B$, were refitted for each $\theta$ to minimize $\chi_{r,\theta}^2$. The set of parameters resulting in the lowest $\chi_{r,\theta}^2$ were selected.

While Sonntag et al. produced proteins with two different deuteration patterns for their SANS experiments, we here focus our analyses on SANS data in which RRM1 was fully deuterated and RRM23 was non-deuterated, since we found this data to be of the highest quality [24]. Therefore, chain labels were included in the PDB with RRM1 being denoted chain A, and RRM2 chain B and RRM3 chain C. This was necessary for subsequent calculations of theoretical SANS curves using Pepsi-SANS [34].

## Tuning protein-water interaction strength in the Martini model

As described in more detail in the results section, we find that simulations using the unperturbed Martini force field yielded structures that were too compact and thus did not fit the experimental SAXS and SANS data. Several atomistic force fields have likewise failed to describe flexible and disordered proteins, but increasing the protein-water interaction strength has in several cases been shown to improve the fit to experimental data [36–38]. Inspired by this work and a previous modification to Martini force field v.2.2 [39] we examined whether a

similar solution could be applied here, and thus varied (increased) the protein-water interaction strength. Specifically, we adjusted the interactions between protein and water by multiplying the $\epsilon$ parameter in the Lennard-Jones potential between water beads and protein beads by a factor $\lambda$ [21] that we varied between 1.0 (unaltered) to 1.5 (50% increase of the protein-water interaction strength). Note, Martini 3 includes "small" and "tiny" beads, along with the normal beads, and the change was also applied to these. The value of $\lambda$ that fitted best with SAXS data was found as the value giving rise to the minimum $\chi_r^2$ when fitting the ensemble-average of the scattering before reweighting. In this case, the parameters $r_0$, or $\Delta\rho$ were taken as the ensemble averages with uniform weights, and $I(0)$ and $B$ were fitted as free (global) parameters.

## SAXS and SANS data

We used one SAXS dataset recorded on non-deuterated TIA-1 and three different SANS datasets with RRM1 fully deuterated (dRRM1) and RRM23 non-deuterated (hRRM23), with SANS obtained in 0%, 42% and 70% $D_2O$ leading to different contrasts of TIA-1 in solution (Fig 1B). All data were collected by Sonntag et al. [24] and obtained from the authors. Varying the deuteration of the solvent gives unique contrast situations in SANS: In 0% $D_2O$ both dRRM1 and hRRM23 have positive contrast but with an inhomogeneous internal contrast due to the higher excess scattering length density, $\Delta\rho$, of dRRM1; in 42% $D_2O$ dRRM1 has positive contrast and hRRM23 has approximately zero contrast; and in 70% $D_2O$ hRRM1 has a negative contrast, and dRRM23 has a positive contrast of similar magnitude. The SAXS data and the SANS data at 0% $D_2O$ and 42% $D_2O$ are included by Sonntag et al. [24], but the SANS data measured at 70% $D_2O$ was measured as part of the original study but not used in their analysis.

## Pair distance distribution functions

We determined the pair distance distribution functions, $p(r)$ (Fig 1C), using Bayesian Indirect Fourier Transformation (BIFT) as implemented in BayesApp [40,41] (available via GenApp [42]). A constant background parameter was used in the transformation, and the distributions were allowed to take negative values. This was necessary in particular for the SANS dataset measured in 70% $D_2O$, as the deuterated domains have positive excess scattering length density (contrast) and the hydrogenated domains have negative excess scattering length density. Such alternating contrasts result in negative values for the $p(r)$ at distances typical for the distance between these domains.

## Combining MD simulations and SAS data by Bayesian reweighting

As mentioned in the introduction, there are several ways to combine MD simulations with SAS data [2–20], but we here used the Bayesian/Maximum Entropy (BME) method [3] and the above-calculated SAXS and SANS intensities to reweight the trajectories. For details of BME see [3] as well as code and examples online https://github.com/KULL-Centre/BME. We refer to recent reviews for an discussion on alternative methods [5,6,43].

The first step in any reweighting approach is to run the MD simulation and compare the average calculated intensity from all frames to the experimental data without any further change of the MD simulations. Discrepancies between calculated scattering and data may arise from insufficient sampling or approximate "forward models" used to calculate scattering data from conformations. Better sampling can be achieved by coarse-graining or other sampling enhancement strategies. Even so, it is often not feasible to achieve full convergence [44], in particular for highly conformationally heterogeneous systems. Moreover, the discrepancies may also be caused by imperfect force fields. By reweighting of the simulated ensemble one can in many cases obtain an ensemble that is consistent with data. In BME reweighting the fit to data

is improved by altering the initial simulation as little as possible [3], in accordance with the principle of maximum entropy. This is ensured by maximizing the relative entropy (the negative Kullbeck-Leibler divergence):

$$S(\boldsymbol{w}) = -\sum_{j=1}^{N} w_j \cdot \log\left(\frac{w_j}{w_j^0}\right),$$

where $w_j^0$ are the initial weights and $w_j$ are the refined weights. The weights are normalized so $\sum_j w_j = 1$, and the initial weights are, in this case, uniformly distributed.

The consistency with data is ensured by minimizing $\chi^2$, defined as usual:

$$\chi^2(\boldsymbol{w}) = \sum_{i=1}^{M} \left(\frac{\sum_{j=1}^{N}(w_j \cdot I_{sim,j,i}) - I_{exp,i}}{\sigma_i}\right)^2,$$

where index $i$ runs over the $M$ measured data points, and index $j$ runs over the $N$ structures in the simulated ensemble.

To balance the two terms, the following expression is minimized:

$$L(\boldsymbol{w}) = \chi^2(\boldsymbol{w})/2 - \theta\, S(\boldsymbol{w}),$$

with $\theta$ being a regularization parameter that balances the trust in data versus simulation [5]. In the applications below we scanned $\theta$ in a range between 1 to 100,000.

The goodness of fit is assessed by the reduced $\chi^2$, defined via the number of degrees of freedom, conventionally estimated as the number of data points, $M$, minus the number of fitting parameters, $K$ (here, $K = 4$ for the four parameters involved in calculating scattering intensities with Pepsi-SAXS or Pepsi-SANS):

$$\chi_r^2 = \chi^2/(M - K).$$

From the relative entropy, the effective fraction of the initial structures used in the final refined ensemble can be estimated as:

$$\phi_{eff} = \exp(S)$$

Expressed in a Bayesian terminology, the prior, i.e. the ensemble from the MD simulations, is updated with the new data, to obtain the posterior, i.e. the ensemble after reweighting. As for all Bayesian methods, the final result is affected by the quality of the prior, as well as the data. In our case, the prior is limited by the accuracy of the force field and the completeness of the sampling. The SAXS and SANS data are intrinsically limited by being low-resolution techniques with a maximal resolution of about 10 Å. The information gain from the data moreover depends highly on the covered $q$-range through the Shannon theorem and the experimental signal-to-noise ratio [45,46]. This implies that the information content of data can be increased by improving data quality e.g. through the counting statistics, and by including more types of data [47], e.g. both SAXS and SANS data. A major focus in the present paper is to examine how and under which conditions SANS can supplement SAXS in such reweighting processes.

## Results

In the first part of the result section we demonstrate how BME reweighting can be used to combine SAXS with MD simulations of flexible proteins, and in the second part we analyse additional information gain of SANS data with different contrasts.

## Part I: Limitations and strength of determining ensembles with the BME reweighting protocol

### An MD simulation with the coarse-grained Martini model does describe the SAXS data accurately

We used MD simulations with the Martini v. 3.0.beta.4.17 coarse-grained model to generate an ensemble of conformations of TIA1. As starting point for our simulations, we built a model using previously determined high-resolution structures of the folded domains (RRM1, RRM2 and RRM3). We applied harmonic restraints within these domains and allowed for full flexibility within the linkers and tails, thus assuming that the three RRMs have similar structures when they are alone or in the the full-length protein. We based this decision on the observation that NMR HSQC spectra of each RRM superposes well with a spectrum of a construct with all three RRMs [25,48] and support it further by noting that e.g. the structure of RRM2 is essentially the same whether in the complex of RNA (PDB ID 5O3J) or in the context of RRM2-RRM3 (PDB: 2MJN).

We performed a 10 μs long MD simulation of TIA1, and examined the consistency between simulation and experiments by comparing the calculated scattering intensity (averaged over all structures in the MD trajectory) with the experimental SAXS data. From this, we observed clear discrepancies, as evident from visual inspection of fit and residuals (Fig 2A). We found that most of the simulated structures had $R_g$ values below the experimentally determined value (Fig 2C) and, that the simulation predicted that TIA-1 is mostly in a collapsed state with $R_g$ of around 20 Å, but with occasional expansions of the three-domain structure, resulting in spikes in the plot of $R_g$ values. Such a compact ensemble is clearly in disagreement with the SAXS data. This observation indicates that the current parameterization of the Martini force field causes too compact structures for the flexible protein TIA-1. We speculate that this may be ascribed to the protein-protein interactions between domains of TIA-1 being too attractive, as such protein "stickiness" has previously been observed for simulations in Martini v2.2 [39,49–51], and Martini v.3.0.beta.3.2 [3]. We considered other reasons for the poor fits, including the fact that we keep domains fixed with elastic networks, and limited accuracy of the calculated SAXS data, but none of these could easily explain the large discrepancy between data and calculated scattering from the unperturbed ensemble.

To improve agreement between simulation and experiment we used the BME reweighting protocol in which the weight of each conformation in the ensemble is modified to improve agreement. By decreasing the parameter $\theta$ (see Methods), it was possible to fit the data more closely (lower $\chi_r^2$), but with a substantial concomitant drop of the effective fraction of frames used (lower $\phi_{eff}$) (Fig 2B). One challenge in BME is to find an appropriate value of $\theta$ [5]. This is most easily found when the $\chi_r^2$ vs. $\phi_{eff}$ curve is convex [3]. In that case, $\theta$ is lowered as long as the decrease in $\chi_r^2$ is substantial, and a value of $\theta$ is chosen, after the curve flattens out, and the decrease in $\phi_{eff}$ is much greater than the decrease in $\chi_r^2$. In the case of the (unmodified) Martini force field, however, the $\chi_r^2$ vs $\phi_{eff}$ curve (Fig 2B) is almost linear. To investigate the effect of the choice of $\theta$, we ran the BME program using different values of $\theta$, and monitored the fit to SAXS data after reweighting, as well as the reweighted distribution of $R_g$ and $D_{13}$ (the distance between domains RRM1 and RRM3), and compared with the non-reweighted distributions (Fig 2C and 2D). For $\theta$ = 5000, the fit was poor with a $\chi_r^2$ of above 40, and a calculated $R_g$ value of 23.3 Å, significantly lower than the experimentally determined value of 27.7 Å. On the other hand, $\phi_{eff}$ at $\theta$ = 5000 was ~21%, so a substantial fraction of the simulation was retained in the reweighted ensemble. At $\theta$ = 150, the fit was seemingly perfect, with a $\chi_r^2$ of unity, and $R_g$ close to the experimental value (Fig 2C). We here note a subtle but important point when comparing the

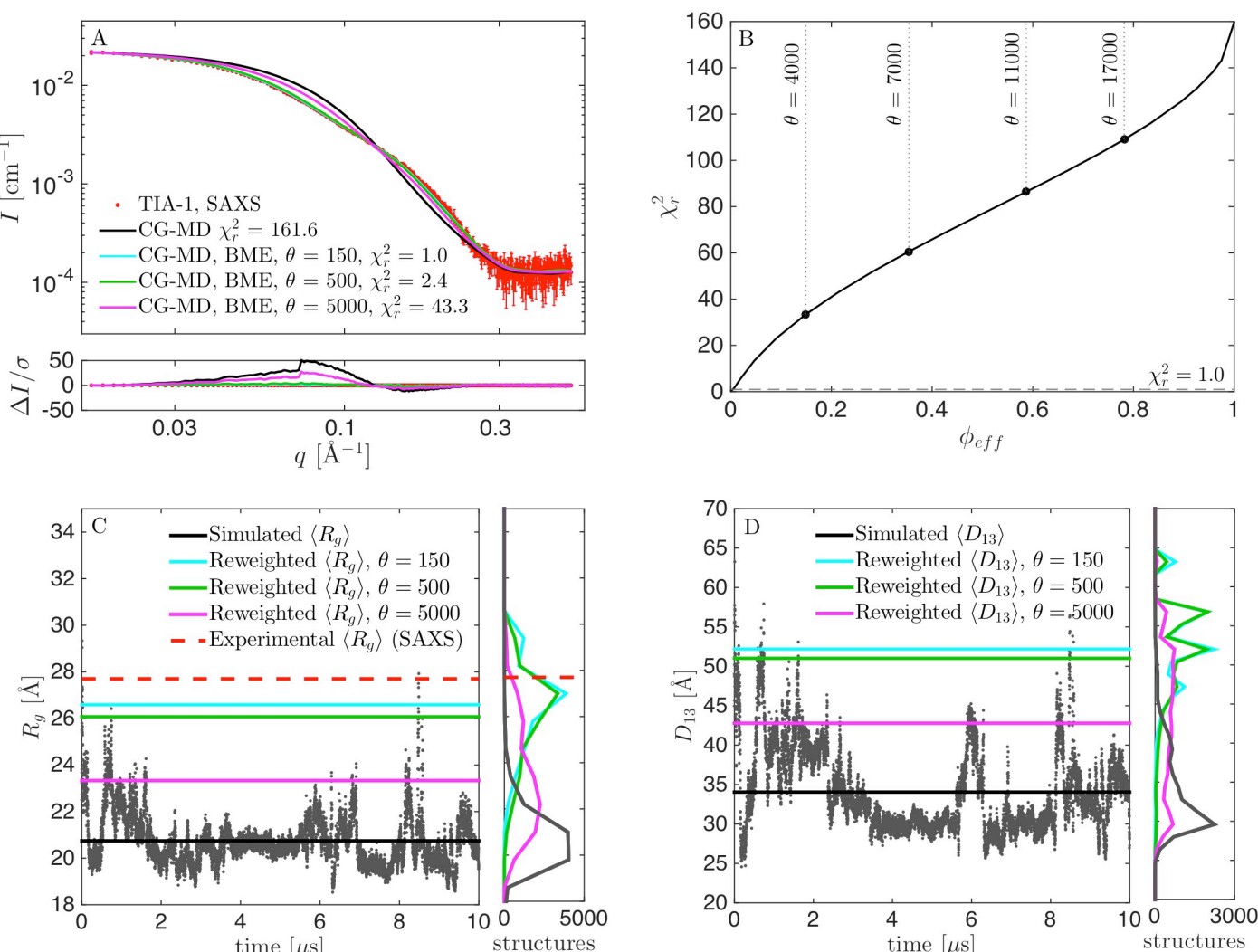

**Fig 2. Results from reweighting the simulation with original force field parameters.** (A) Fit to SAXS data, before (black) and after reweighting at $\theta = 150$ (cyan), $\theta = 500$ (green), and $\theta = 5000$ (purple). (B) $\chi_r^2$ vs. $\phi_{eff}$ for selection of $\theta$. (C) Calculated $R_g$ during the simulation, including mean value (black), experimental $R_g$ from SAXS (red) and mean $R_g$ from the reweighted ensembles (green). Corresponding histograms in the right panel. (D) Calculated $D_{13}$ before and after reweighting.

simulations to the experimental SAXS data. Specifically, even for a perfect ensemble we do not expect perfect agreement between the $R_g$ from SAXS and the $R_g$ calculated from the atomic coordinates in the simulations, since the former is based on an experimental determination in water and include the hydration shell around the protein, while the latter is calculated from the protein structure in vacuum. Instead, all our quantitative comparisons between the simulations and experiments are based on calculating the scattering data, $I(q)$, from the conformational ensembles, including modelling the contribution of the solvent layer to the scattering intensity, enabling us to compare the simulations directly with the data rather than with derived quantities such as the $R_g$. Thus, after reweighting with $\theta = 150$ the consistency with SAXS data is excellent. However, $\phi_{eff}$ was only about 0.4%, i.e. just 40 out of the 10,000 frames effectively contributed to the reweighted ensemble. We concluded that this value of $\theta$ was too low, and found instead that $\theta$ around 500 was a better compromise, since the fit, as judged also by visual inspection, was almost as good as for $\theta = 150$, but $\phi_{eff}$ was increased 10-fold to ~1%. That is, a good fit could be obtained, but substantial reweighting was necessary.

## Altering the Martini force field by increasing the protein-water interaction strength

To improve the fit between the ensemble from the MD simulations and SAXS, we explored a rescaling of the protein-water interaction strength similar to what has previously been done for all-atomic force fields [36–38] and Martini v.2.2 [39,50,51]. By changing the solvent properties towards a better solvent (i.e. increasing protein-water interactions relative to protein-protein interactions), we stabilized structures with increased expansion and solvent accessible surface, and thus expected these to be visited more frequently in the simulations.

We changed the protein-water interaction strength by a factor $\lambda$ in the range from 1.00 (unaltered) to 1.50 (50% increase of the interaction strength) and monitored the calculated averaged $R_g$ from the simulations, as well as the fit of calculated intensities to the experimental SAXS and SANS data (Fig 3). As described above, the fit to SAXS data before reweighting was poor at $\lambda = 1.00$, as assessed by visual inspection and a $\chi^2_r$ of above 40. However, the fit dramatically improved as $\lambda$ increased, up to a value of $\lambda = 1.06$, where a very good fit was achieved with a $\chi^2_r$ of 2.8 (Fig 3). When $\lambda$ was increased beyond that point, the fit again worsened, and $R_g$ also increased to values above the SAXS-estimated $R_g$ indicating that the protein structures were generally too extended above $\lambda = 1.06$. In other words, the solvent became too good. We note that the fit seemingly got worse in the first step from $\lambda$ 1.00 to $\lambda = 1.01$, before improving again. This is likely due to difficulties in converging at low $\lambda$ values because of stickiness between the domains.

Calculating the $R_g$ in the simulation with $\lambda = 1.06$ revealed mostly expanded structures with $R_g$ values up to ~45 Å, but also some more collapsed forms with $R_g$ of 20–25 Å (Fig 4). We decided to use this simulation as a prior for further reweighting (Fig 4B), which lead to a very good fit to SAXS data with $\chi^2_r$ of 1.0, and $\phi_{eff}$ of 83% (Fig 4B and 4C). Both the $R_g$ (Fig 4C) and the distance between RRM1 and RRM3 ($D_{13}$; Fig 4D) changed only little due to reweighting.

## Fitting the data twice

The above protocol included two fitting steps: First, we fitted $\lambda$ by a grid-search to find the value that best matched SAXS data. Second, we reweighed the trajectory weights to obtain even better consistency with data. That is, the prior in the BME protocol [3], was not a true prior, as the initial weights, which were input in the reweighting protocol, had already been adjusted against data. Optimally, in a Bayesian framework, only one fitting step should be applied to obtain consistency between simulations and a given experimental dataset. However, the two-step fitting protocol here provided the most reliable results. We suggest that such a two-step fitting protocol is necessary when the force fields leads to a poor initial consistency with data, i.e. that all relevant states has not been sampled. For TIA-1, the extended states were not sufficiently sampled with the pure Martini force field. We note that the reweighting only changed the distributions for $R_g$ and $D_{13}$ slightly (Fig 4C and 4D), so in this case, the first fitting step where $\lambda$ was adjusted, was sufficient to obtain a reliable ensemble. However, it is not generally the case that adjusting a single parameter in the force field leads to consistency with experimental data.

## Reweighting after simulating an ensemble with suboptimal force fields

In the case described here we could tune a single parameter in the force field to obtain a good match between experiment and simulation. This was possible as the elastic network keeps the structure of the domains rigid, so tuning of $\lambda$ did not affect the structure of the domains. This would not be the case for an all-atomic simulation, and hence one would in general have to

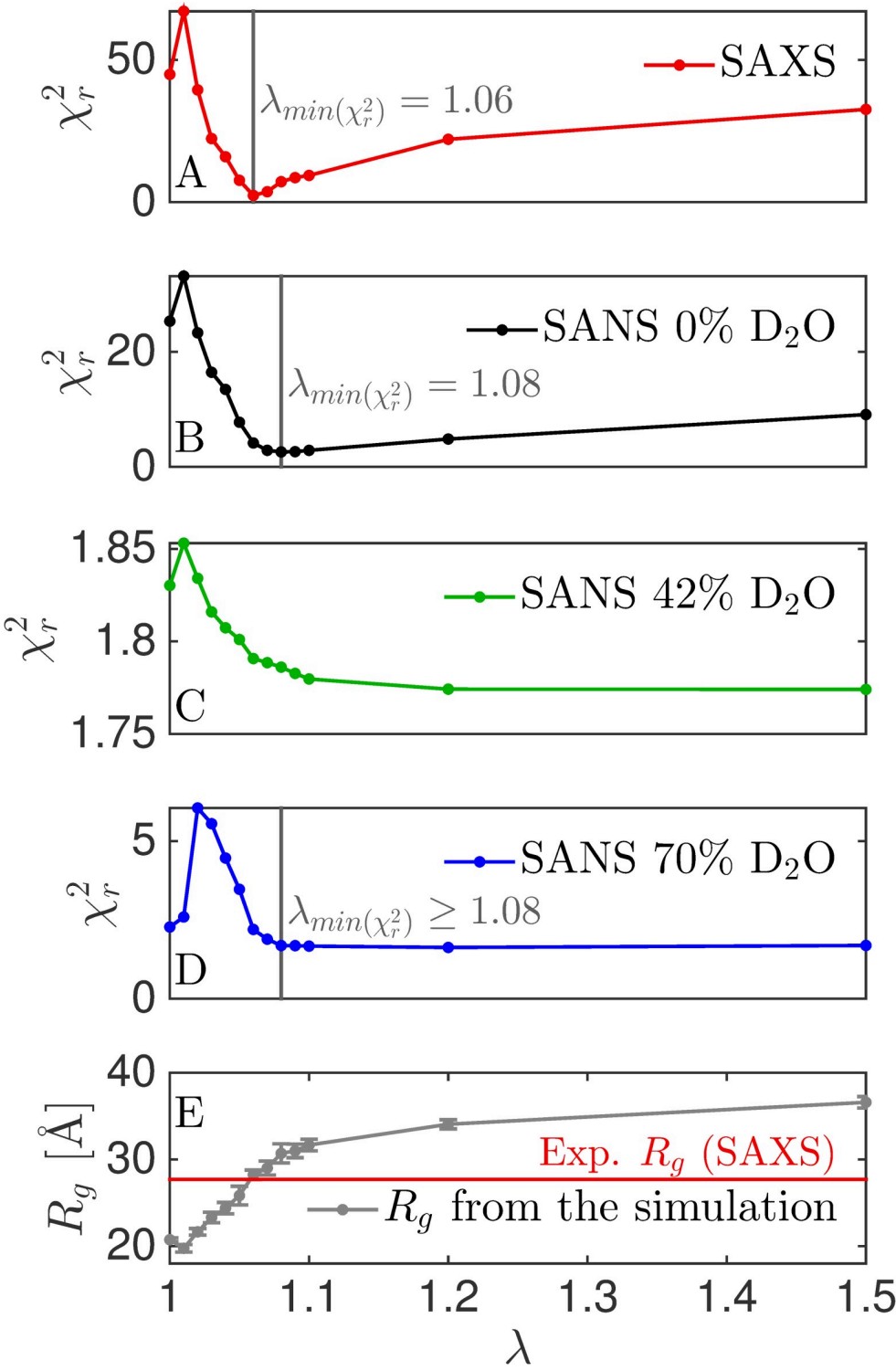

**Fig 3. Tuning the protein-water interaction strength of the Martini force field.** We varied the protein-water interaction strength by a factor $\lambda$, and calculated $\chi_r^2$ for each dataset (A-D). Vertical grey lines are the values of $\lambda$ giving the best fit to the given dataset. No vertical line is given for SANS at 42% D2O as the variation is $\chi_r^2$ is very small. (E) Average $R_g$ as calculated directly from the simulation. The horizontal red line is the value of $R_g$ determined from the SAXS data.

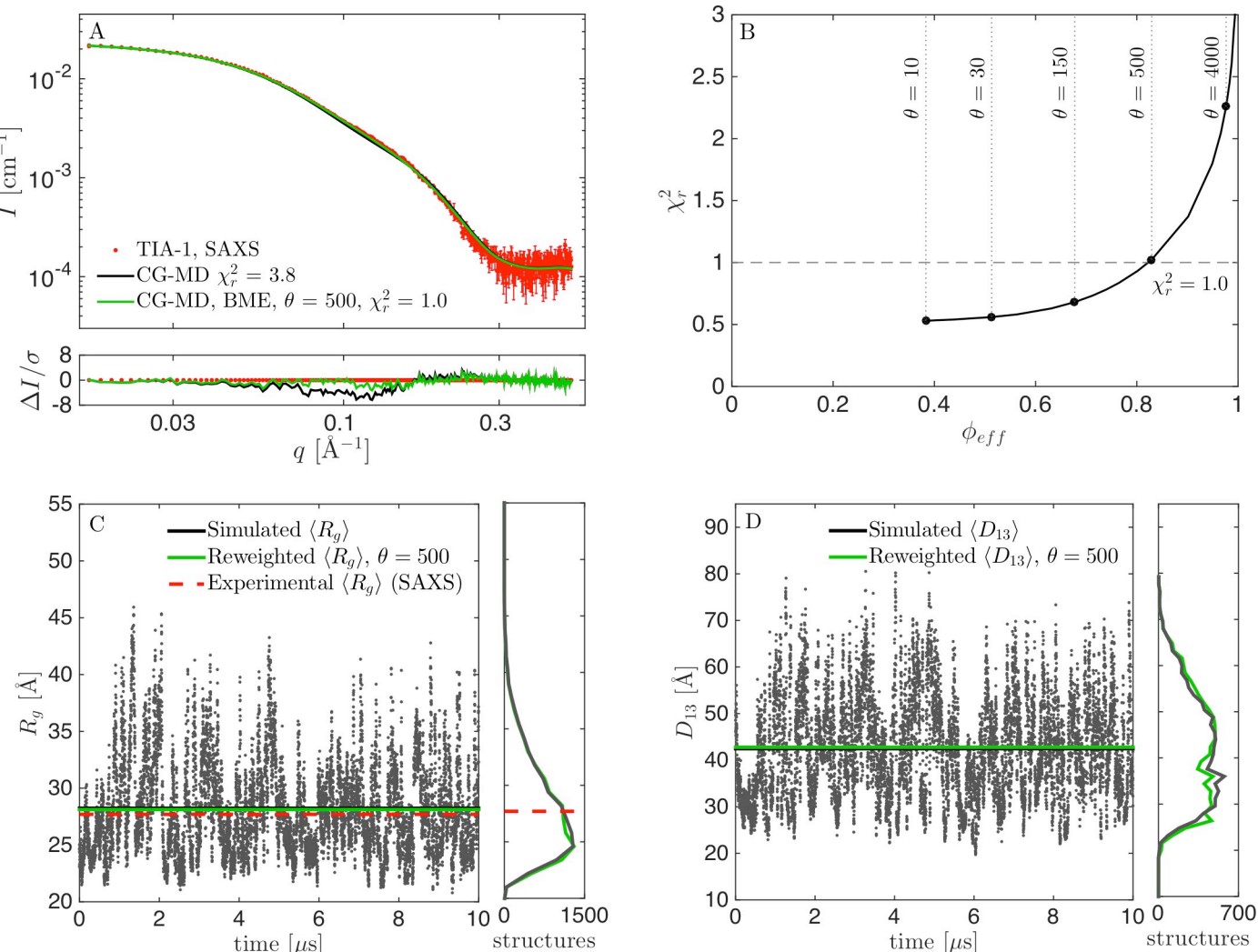

**Fig 4. Results from reweighting the simulation with the optimized force field ($\lambda = 1.06$).** (A) Fit to SAXS data with adjusted force fields before (black) and after reweighting at $\theta = 500$ (green). (B) $\chi_r^2$ vs. $\phi_{eff}$ for selection of $\theta$. (C) $R_g$ calculated from structures during the simulation, including the mean value (black), experimental $R_g$ from SAXS (red), and mean $R_g$ from the reweighted ensemble (green). Corresponding histograms in the right panel. (D) Calculated $D_{13}$ before and after reweighting.

work more to balance parameters in a given force field/prior. Therefore, in the general case one would need to be able to reweight an ensemble even if it has been performed with a force field (prior) that leads to substantial deviations from experiments. An interesting question is therefore, how the reweighted ensemble depends on the prior, i.e. what happens when reweighting from an unaltered ($\lambda = 1.0$), a slightly improved (e.g. at $\lambda = 1.04$) and a close to optimal ($\lambda = 1.06$) prior. We note that the altered force fields are improved for this specific system only, and may perform worse for other systems.

For the suboptimal prior at $\lambda = 1.04$, the fit to SAXS data could be significantly improved by reweighting, and we achieved a $\chi_r^2$ of 1.0 by reweighting to $\phi_{eff} = 56\%$ (Fig 5). Before reweighting, TIA-1 was occasionally in a collapsed state with $R_g$ of 20–25 Å, but most of the time in more expanded states with $R_g$ up to ~40 Å (Fig 5C). Most structures had calculated $R_g$ below the experimental value, and the average value was underestimated. However, a considerable amount of structures with larger $R_g$ ensured that reweighting could be successfully applied.

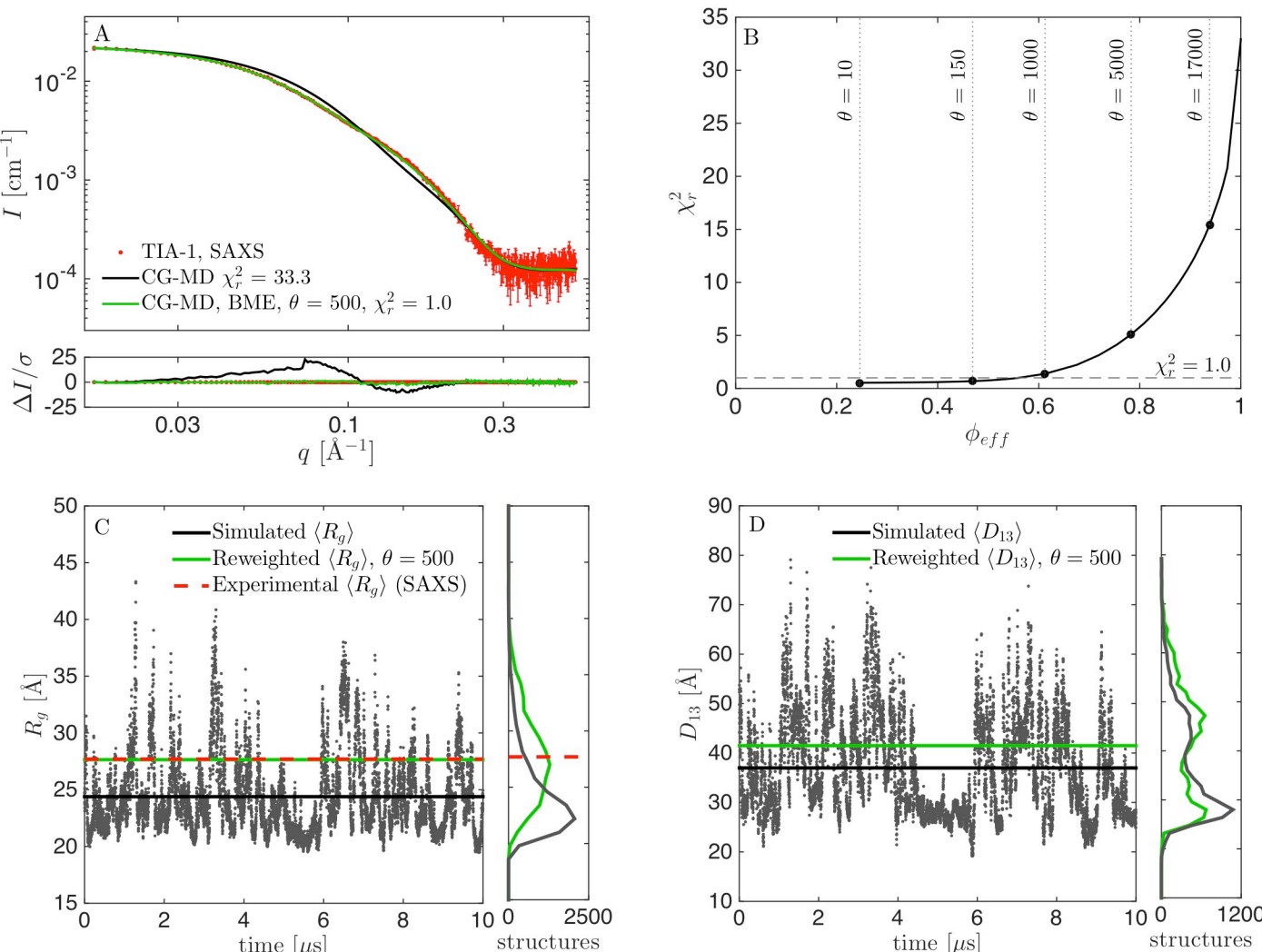

**Fig 5. Results from reweighting the simulation with underestimated protein-water interaction strength ($\lambda = 1.04$).** (A) Fit to SAXS data with adjusted force field before (black) and after reweighting at $\theta = 500$ (green). (B) $\chi_r^2$ vs. $\phi_{eff}$ for a selection of $\theta$. (C) $R_g$ calculated from structures during the simulation, including the mean value (black), experimental $R_g$ from SAXS (red), and mean $R_g$ from the reweighted ensemble (green). Corresponding histograms in the right panel. (D) Calculated $D_{13}$ before and after reweighting.

The distribution for $R_g$ shifted significantly after reweighting, but, in contrast to the distribution for $R_g$ at $\lambda = 1.0$ (Fig 2C), there was still a large overlap between the initial and the reweighted distributions, which was also reflected in the much higher value of $\phi_{eff}$.

We directly compared the reweighted ensembles from $\lambda = 1.0$ (Fig 2), $\lambda = 1.04$ (Fig 5) and $\lambda = 1.06$ (Fig 4), as well as additional ensembles generated with $\lambda = 1.08$ (S2 Fig) and 1.10 (S3 Fig), by examining the resulting distributions for $R_g$ and $D_{13}$ (Fig 6). Optimally, the distributions after reweighting should resemble that from reweighting of the best force field ($\lambda = 1.06$). The reweighted distributions from the unaltered force field ($\lambda = 1.00$) differed markedly from the rest (Fig 6). The reweighted distributions for the good and suboptimal force fields, on the other hand, were rather consistent, showing that reweighting can be used whenever the force field is "good enough". An obvious question is then what "good enough" means. First, the distributions of some central parameters can be compared before and after reweighting. A large

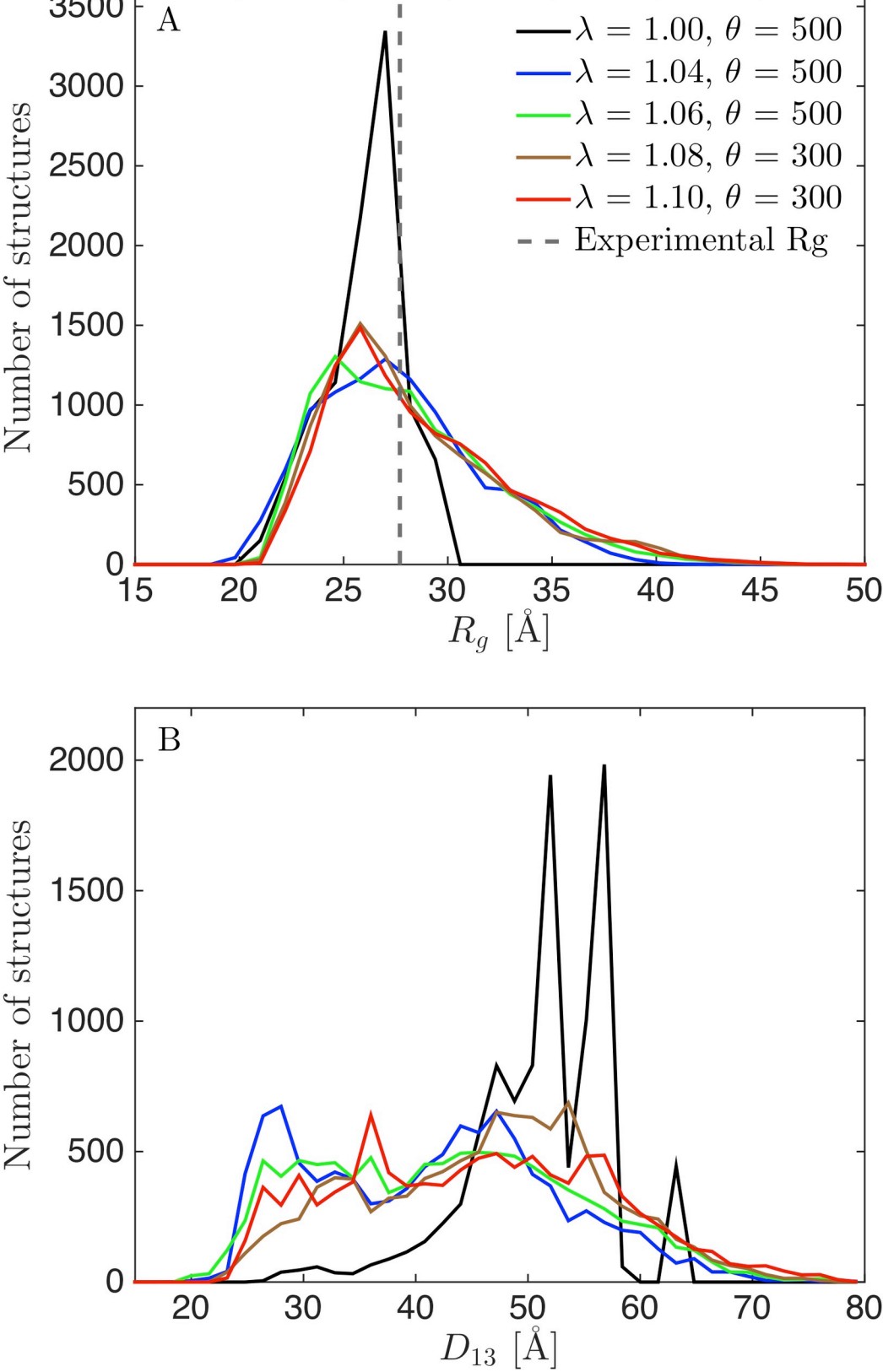

**Fig 6. Distributions of $R_g$ (A) and $D_{13}$ (B) after reweighting.** Reweighted from simulations using $\lambda$ = 1.00 (black), 1.04 (blue), 1.06 (green), 1.08 (brown) and 1.10 (red). Experimental $R_g$ given in (A) as a dotted line.

overlap (Figs 4 and 5) indicates that the force field is good enough, whereas a small overlap (Fig 2) indicates the opposite. Second, the value of $\phi_{eff}$ is a good indicator. For the unaltered force field, we needed to heavily reweight the ensemble to obtain consistency with data, with a $\phi_{eff}$ of about 1%. For the other force fields, $\phi_{eff}$ varied between 36% ($\lambda = 1.10$) and 83% ($\lambda = 1.06$). We conclude that the most reliable results could be obtained by altering the force field before reweighting to obtain a force field that was "good enough". However, we also note that the ensemble obtained with the unaltered force field ($\lambda = 1.00$) was still considerably improved by reweighting.

## Part II: Information gain from the inclusion of SANS data

In the results described above we used coarse-grained simulations and SAXS data to study the conformational ensemble of the three-domain protein TIA-1. Because SAXS experiments are generally sensitive to the overall distribution of mass within the protein, such experiments may not be able to distinguish, for example, fluctuations of the distance between RRM1 and RRM2 *vs.* between RRM2 and RRM3. Using selective protein deuteration and SANS, scattering from specific domains can, however, be highlighted or dampened. Thus, SANS and contrast variation provide additional information to the SAXS data. The question is how much extra information can be gained considering that SANS data are generally noisier than SAXS data? In other words, how much are the final reweighted distributions altered by inclusion of SANS data?

### SANS data at different contrast situations carry different structural information

We used data from three SANS contrasts in the present study, all measured on constructs from segmental labelling with RRM1 deuterated and RRM23 hydrogenated. The samples were measured in respectively 0%, 42% and 70% $D_2O$. The sample in 0% $D_2O$ contained scattering contributions from both RRM1 and RRM23, but with RRM1 having significantly higher excess scattering length density. At 42% $D_2O$, RRM23 was matched out, and the scattering signal originated solely from the deuterated RRM1 domain. At 70% $D_2O$, RRM1 and RRM23 had respectively positive and negative excess scattering length densities (contrasts). This is evident from the $p(r)$ function (Fig 1C) with alternating sign, which can only appear if parts of the sample have contrast with opposing signs. This contrast is generally considered attractive because it theoretically contains important information about the internal structure of the protein. It, however, has forward scattering, *I(0)*, close to zero and generally low scattering intensity in the full $q$-range. This unfortunately results in a low experimental signal-to-noise ratio, as clear from the relatively larger experimental errors (Fig 1B) and hence in less information rich SANS data in practice.

### Different optimal values of $\lambda$ found with SANS than that found with SAXS

Similar to our analysis of SAXS data, we determined the fit to the SANS data for simulations with varying values of the protein-water interaction strength (Fig 3). Interestingly, the best fit to the SANS data at 0% $D_2O$ was at $\lambda \sim 1.08$ and for SANS at 70% $D_2O$ it was at $\lambda \geq 1.08$. The SANS data at 42% $D_2O$ were fitted best with the highest tested protein-water interaction strength of 1.50, but the fit was relatively good at all values ($\chi_r^2 < 2$). As SANS at 42% $D_2O$ only "sees" the RRM1 domain, the result may indicate that this domain is slightly too compact in the simulation. When optimizing $\lambda$ against all SANS data, a value of about 1.08 was optimal, i.e. slightly higher than the value of 1.06 found with SAXS alone. This difference from SAXS

data may stem from difference in contrast situation, but could also be an effect of $D_2O$ being a different solvent than $H_2O$, such that the samples for SAXS and SANS are not structurally fully identical. We note, however, that the agreement with the SAXS and SANS obtained with these two different values of $\lambda$ were rather similar (Fig 3).

## Inclusion of SANS data had only limited effect on reweighted distributions

We proceeded to examine the information in the SANS data by reweighting the simulations. We reweighted from the simulations with the unaltered force field ($\lambda = 1.00$), to monitor the largest effect of the reweighting protocol. In particular, we reweighted with either the SAXS data, with each of the SANS datasets, or with all data simultaneously and calculated the distributions of $R_g$ and $D_{13}$ (Fig 7). The distributions after reweighting were almost identical when using, respectively, SAXS alone and SANS at 0% $D_2O$ alone. Due to the contrast match-out of the hydrogenated RRM2 and RRM3 domains, the SANS data at 42% $D_2O$ had only limited information about the overall structure of TIA-1, and thus reweighting with this data alone only shifted the distributions marginally. The SANS dataset at 70% will in principle contain information about the whole complex, but due to the low signal-to-noise ratio, this dataset alone was not sufficient to shift the distribution as much as the SAXS data or the SANS data obtained at 0% $D_2O$. When including all data, the final distributions reflected a mix of the distributions obtained by using each of the datasets separately. The distribution obtained after including all data was, however, qualitatively similar to the distribution obtained after reweighting with SAXS data alone, albeit slightly closer to the initial distribution.

   Those results indicate that, for this specific system and data, the SANS data add only limited extra information about the distributions of $R_g$ and $D_{13}$ when high-quality SAXS data is already available. We found that inclusion of the SANS data resulted in a slightly more conservative distribution, i.e. one that is closer to the initial distribution. This was likely because further reweighting did not improve the fit to SANS data at 70% $D_2O$, despite improving the fit to SAXS data and SANS data at 0% $D_2O$. Given the similar results when reweighting against SAXS data and SANS data measured at 0% $D_2O$, the latter could in principle be used instead of SAXS data. We note, however, that SAXS instruments are generally more available, have higher flux and need less sample, so SAXS is in most cases the first method of choice. The SANS data at 42% on the other hand probes mostly the deuterated RRM1 domain, since RRM23 are matched out in this experiment. The results from this experiment was fully consistent with what we already know from NMR on RRM1, and does not provide additional information about the overall structure and interdomain flexibility of TIA-1. But the contrast could potentially be highly relevant when studying e.g. how one protein changes shape under influence of other (matched-out) proteins or RNA molecules, or if the domain had actually changed conformation. The 70% SANS contrast is particularly relevant for protein/RNA complexes (the original study included also SANS data on the complex between TIA-1 and an RNA molecule [24]), as RNA is nearly matched out in 70% $D_2O$, and as discussed above it contains, in principle, very useful information about the overall structure, but due to the low signal-to-noise ratio it only provided limited information on the overall flexibility of TIA-1.

## SANS used for cross-validation and determination of $\theta$

The SANS data can also be used to cross-validate the reweighting of SAXS data to prevent overfitting [7], and estimate the best value of $\theta$, which quantifies the trust in the MD simulation [5]. We thus reweighted trajectories generated using $\lambda$ of 1.00 and 1.06 using SAXS data, and monitored the effect of $\chi_r^2$ calculated using SANS data for cross-validation (Fig 8). At $\lambda = 1.00$, the SANS data at 42% $D_2O$ fitted (as expected) equally well for all values of $\theta$, whereas the

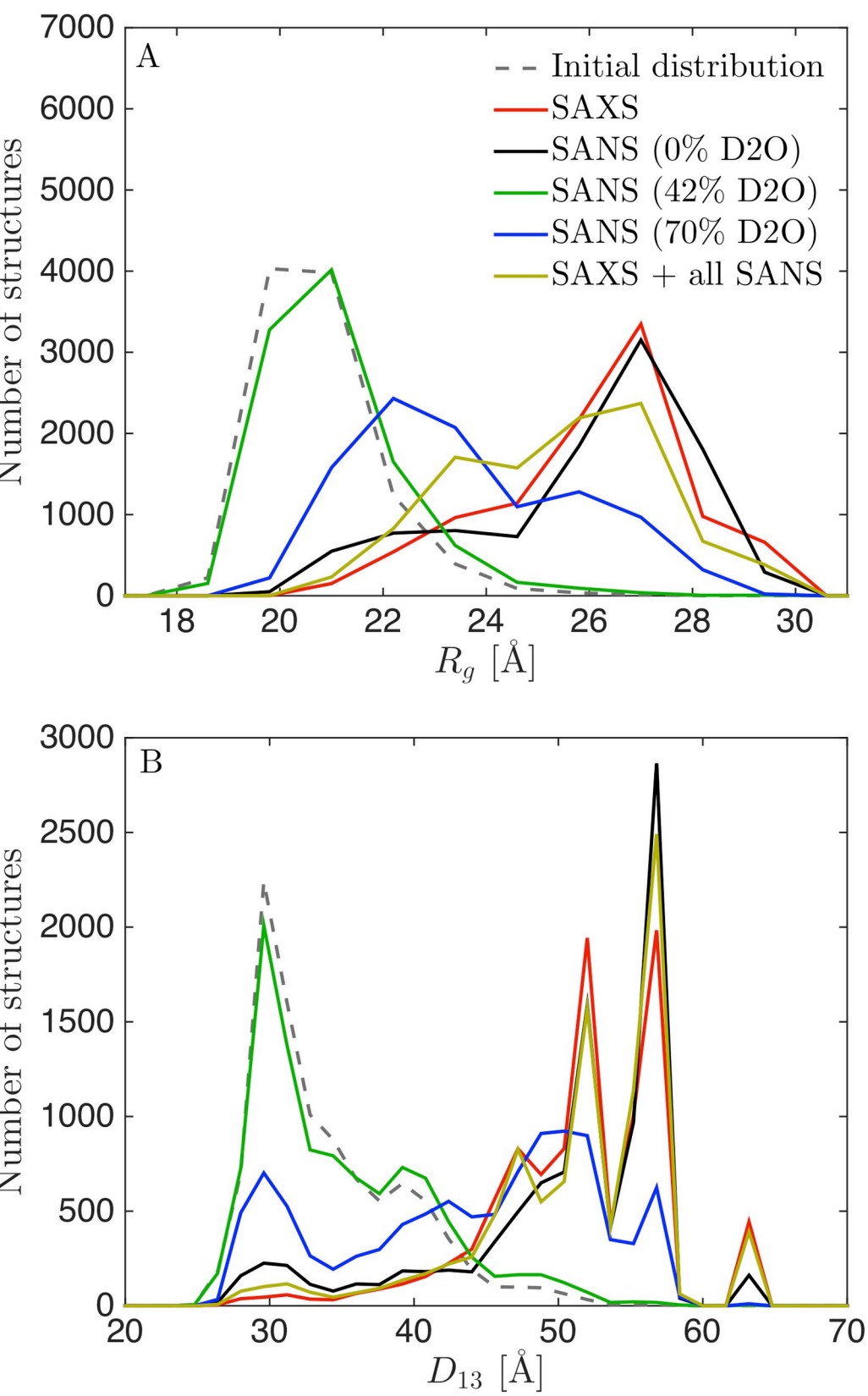

**Fig 7. Distributions of $R_g$ (A) and $D_{13}$(B) after reweighting from unaltered simulation ($\lambda$ = 1.00) using SAXS and SANS data.** (A) Distribution of $R_g$ and (B) distribution of $D_{13}$ after reweighting with SAXS alone (red), with SANS at 70% $D_2O$ (black), SANS at 42% $D_2O$ (green), SANS at 70% $D_2O$ (blue) and with all SAXS and SANS data (yellow).

fit to the two other SANS contrasts were improved along with the improvement to the fit of SAXS data. That is, the SANS data at 0% and 70% $D_2O$, when combined, carry rather similar structural information as that contained in the SAXS data. Starting with the simulation that best fits with SAXS data ($\lambda$ = 1.06), however, gave a more subtle picture. The SANS at 42% $D_2O$ was again fitted equally well for all values of $\theta$, and this time that was also the case for SANS data at 70% $D_2O$. However, the agreement with the SANS data at 0% $D_2O$ worsened slightly as the fit to SAXS data improved. This result illustrates that the SANS data contain some structural information not captured fully by the SAXS data, though additional experiments (such as SAXS measurements of the deuterated samples and in $D_2O$) would be useful to determine whether these small differences come from differences in sample conditions. The additional structural information was also reflected in the different optimal values of $\lambda$ found in SANS and SAXS (Fig 3).

For $\lambda$ = 1.04, the agreement with the SANS datasets improved as $\theta$ decreased until around $\theta$ = 1000, where the fit to SANS at 0% $D_2O$ and at 70% $D_2O$ slowly worsened (S4 Fig). Not surprisingly, SANS at 42% again fitted well for all value of $\theta$. Reweighting the simulation at $\lambda$ = 1.08 showed the same picture as for $\lambda$ = 1.06, namely that improving the fits to the SAXS data slightly worsened the fit to the SANS data at 0% $D_2O$ (S4 Fig).

Thus, overall our results suggest that the SANS and SAXS data provide similar information when the initial simulations are far away from the "correct" ensemble. As the simulated ensemble gets closer to the final ensemble obtained from fitting both $\lambda$ and reweighting against the SAXS data, then we find that the SANS data contains a small amount of extra information.

## Optimal SANS contrasts

As discussed above, although the SAXS and SANS data are overall consistent, there is some additional information to be gained from the SANS data. The 42% $D_2O$ contrast gives information about the RRM1 domain structure, and the data appears to be relatively accurately described by the NMR structure. The 0% and 70% $D_2O$ contrasts pointed towards higher values of $\lambda$, i.e. towards more extended structures, than the SAXS data alone. However, although this observation suggested some orthogonal information, the SANS data only modestly altered the final ensembles (after reweighting). This was on one side because SANS data at 0% $D_2O$ and 70% $D_2O$, where all domains were "visible" (i.e. not matched out), carried much of the same information as the SAXS data, namely information about the overall structure of TIA-1. Also, SANS data generally had lower signal-to-noise ratio and more limited $q$-range than SAXS data, and thus contained less structural information [45,46].

To potentially gain more information from additional SANS contrast, it is worth discussing the optimal SANS conditions, and what could in principle be gained from them. There are two major points to be aware of when selecting SANS contrasts in this case. First, SAXS carries information about the bulk contrast, i.e. where all domains add to the SAXS signal simultaneously (Fig 9A). An optimal SANS contrast should therefore avoid such bulk contrast situations to be complementary to the SAXS data. The second point is the signal-to-noise ratio. An effective way to increase the signal-to-noise ratio in SANS is to minimize the incoherent background scattering from $H_2O$ in the sample. A relevant contrast situation is therefore obtained at 100% $D_2O$, where the signal-to-noise ratio can be improved radically, and data quality comparable with SAXS data can be obtained, even for challenging protein systems that are difficult

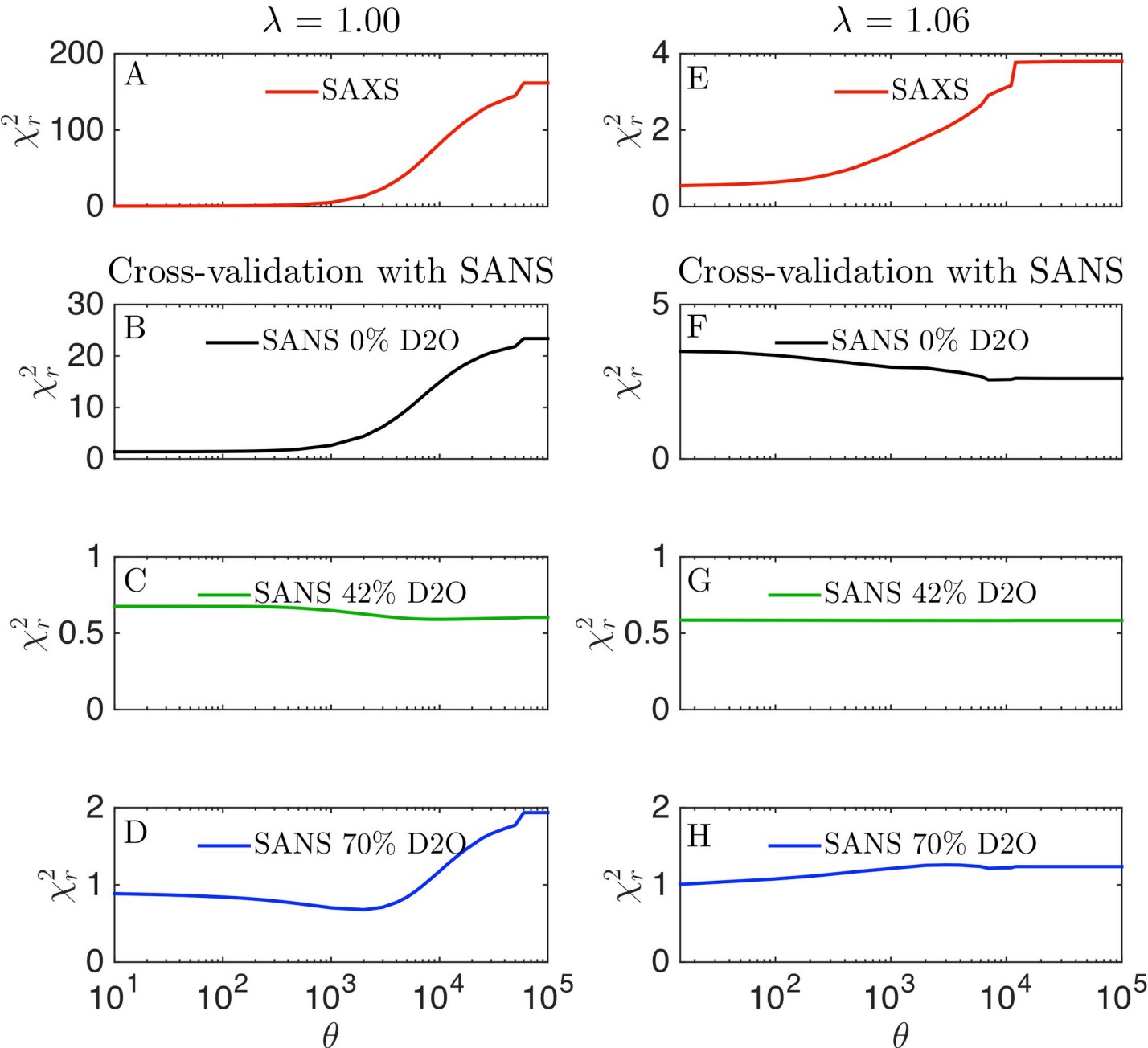

**Fig 8. Cross-validation with SANS data.** Simulations with either (A–D) $\lambda$ = 1.00 or (E–H) $\lambda$ = 1.06 were reweighted against SAXS data at several different values of $\theta$. Agreement with (A, E) SAXS data after fitting to the SAXS data (red), or cross-validated with SANS at (B, F) 0% $D_2O$ (black), (C, G) 42% $D_2O$ (green), (D, H) 70% $D_2O$ (blue).

to express in large quantities [52]. Moreover, to complement the SAXS, one out of three domains should ideally be matched out, in order to have a contrast situation where only two domains contribute to the total scattering (Fig 9). As in the original study, this can be obtained by partial deuteration of one of the domains [53], and assembly by sortase [24]. In practice, only two of these contrast are easily feasible, as the combination with RRM2 being deuterated and RRM1 and RRM3 being hydrogenated (Fig 9C) requires two ligation steps in the sortase protocol [24].

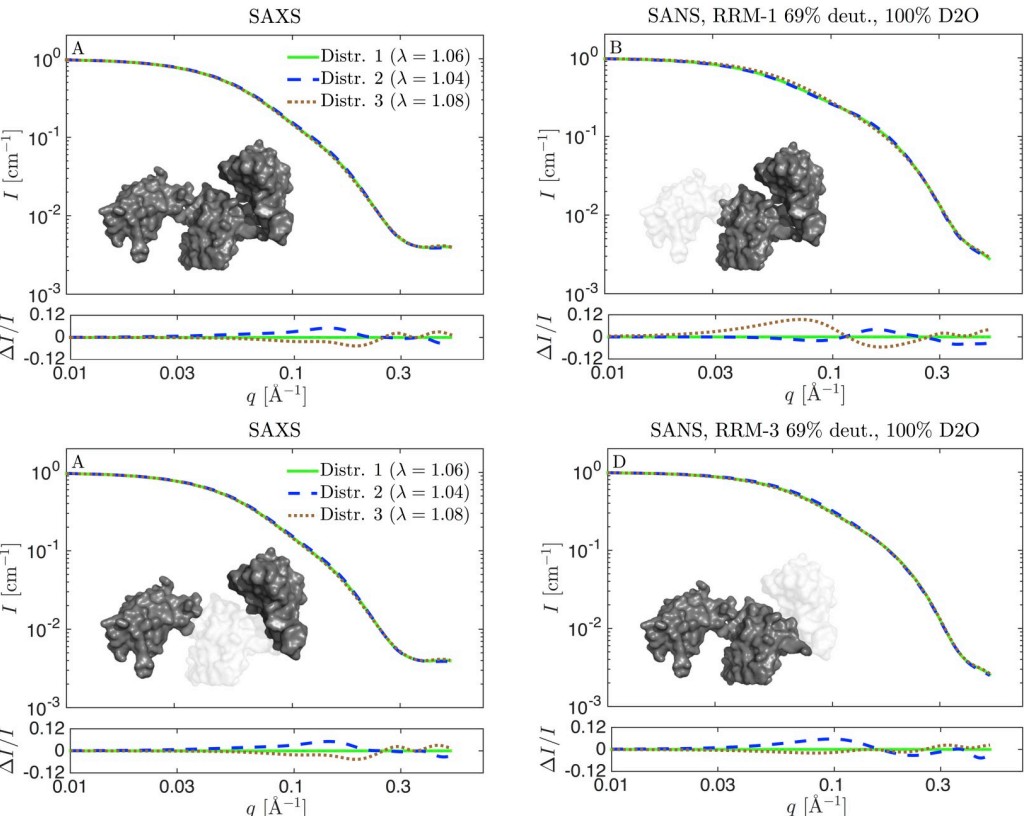

**Fig 9. Theoretical SAXS and SANS scattering for reweighted ensembles in Fig 6.** Distribution 1 is the reweighted ensemble from the simulation with $\lambda$ = 1.06 (green), distribution 2 is from reweighted ensemble from the $\lambda$ = 1.04 simulation (blue), and distribution 3 is from the $\lambda$ = 1.08 simulation (brown). (A) SAXS data. (B-D) SANS calculated with respectively RRM1, RRM2, or RRM3 matched out by perdeuteration to 69% and measured in 100% $D_2O$. Residuals show the relative difference to the scattering from the distribution reweighted from the simulation with $\lambda$ = 1.06.

To investigate the possible information gained from these "optimal SANS data" in 100% $D_2O$ and with one domain matched out, we calculated the theoretical SANS scattering of three of the distributions in Fig 6. The chosen distributions were qualitatively similar, and came from reweighting with SAXS data starting from simulations with respectively $\lambda$ = 1.04, 1.06 and 1.08 force fields. The reweighted ensembles gave equally good fits to SAXS data, as assessed by the $\chi_r^2$ close to unity. However, there were some minor differences in the underlying ensembles, and the question was whether one with optimally chosen SANS contrast would realistically be able to probe these differences. We use the notation distribution 1 (reweighted from the simulation with $\lambda$ = 1.06), distribution 2 ($\lambda$ = 1.04), and distribution 3 ($\lambda$ = 1.08).

As expected and per design, the theoretical SAXS data for the three distributions were very similar, with only small differences in the residuals (Fig 9A). In the residuals, each theoretical curve was compared to the one calculated from distribution 1, and divided by the intensity, to obtain the relative residuals. For SANS with RRM1 matched out (Fig 9B), distribution 3 gave slightly different scattering, and might be discriminated from the two others if very good SANS data from a contrast with RRM1 matched out were available. Hence, this contrast is optimal for mapping out the distribution for the distance between domain RRM2 and RRM3. The sample with RRM2 matched out (Fig 9C) would, as mentioned above, be the most challenging sample to prepare. It did however show a significant difference between the scattering distribution 3 and the two others. This contrast is an optimal choice for measuring the distance

between RRM1 and RRM3, $D_{13}$. In the last contrast situation, RRM3 is matched out (Fig 9D). Here, distribution 2 could best be distinguished from the others, but the relative differences were rather small and the experiment would have to be optimized to a great extent to make the distinction. It is the best SANS contrast for measuring the distance between RRM1 and RRM2. Thus, if SANS data with one or more of these contrast situations were available with a sufficiently good signal-to-noise ratio, the distributions for $R_g$ and the inter-domain distances could have been determined more precisely, but the overall structural conclusions would not be altered significantly. For such subtle differences to be useful in a reweighting protocol, the forward model used to calculate the scattering from coordinates also has to be very precise. In the present case we expect that a more detailed back-mapping protocol might be necessary (see Methods and S1 Fig).

## Discussion

### Reweighting simulations with different force fields

We have shown that an ensemble for TIA-1 obtained from simulations with the latest Martini coarse-grained force field did not match well SAXS and SANS data. We, however, were able to obtain ensembles that fitted the data much better by strengthening the protein-water interactions through an adjustment of the force field. The best fit to the SAXS data was obtained at $\lambda$ = 1.06, i.e. with about 6% increase of the protein-water interaction strength. For SANS data, a slightly higher value for the protein-water interaction strength best fitted the data. Further work on additional proteins is needed to assess whether such 6–8% increase of the protein-water interaction strength is also applicable to other systems simulated with the Martini force field. Although perhaps fortuitous, the rescaling is similar in magnitude to the adjustments seen for some all-atom force field adjusted to simulate proteins with intrinsically disordered regions [36–38]. For the TIA-1 system, reliable distribution for collective variables such as $R_g$ and $D_{13}$ could be obtained from several non-perfect force fields by reweighting against experiments. For these force fields ($\lambda$ = 1.04–1.10), the reweighted distributions were qualitatively the same as the reweighted distributions obtained from the best force field ($\lambda$ = 1.06). This was also reflected in the obtained values for $\phi_{eff}$. For $\lambda$ =1.00 (the unaltered force field), the trajectory needed to be heavily reweighted to a point where $\phi_{eff}$ was less than 1% before a good fit was achieved. For the force fields that proved to be "good enough", $\phi_{eff}$ after reweighting to SAXS data varied between 36% at $\lambda$ = 1.10 (S2 Fig), and 83% at $\lambda$ = 1.06 (Fig 4). A good force field can be recognised by a substantial overlap between the distribution for collective variable before and after reweighting. This was not the case for the simulation at $\lambda$ = 1.00 (Fig 2), but was the case e.g. for $\lambda$ = 1.04 (Fig 5), and $\lambda$ = 1.06 (Fig 4). For other systems it might be difficult to determine what the important collective variables are, and in those cases, $\phi_{eff}$ may still be utilized to assess the quality of the force field against experimental data, as it measures overlap between prior and reweighted distribution independently of any choice of parameter.

The value of $\phi_{eff}$ relates to how much the force field needs to be modified so that the ensemble agrees with the data. This, in turn suggests that $\phi_{eff}$ may quantitatively relate to the error of the force field. Indeed, it can be shown that the relative entropy, $S$, is proportional to the free energy difference between two ensembles [54]. Using this formalism and the values of $\phi_{eff}$ needed to reweight the ensembles sampled at $\lambda$ = 1.0, 1.04, 1.06 and 1.10 to a $\chi_r^2 = 1$ ($\phi_{eff}$ = 0.4%, 56%, 86% and 36%, respectively) we find that that the force field errors are 5.5 $k_BT$, 0.6 $k_BT$, 0.1 $k_BT$ and 1.0 $k_BT$, respectively. The exact interpretation of these estimates is, however, complicated by the fact that this only says something about how wrong the force field is, as viewed from the SAXS data, so that the force field could still be wrong even if $\phi_{eff}$ is high. We

thus suggest that more systems need to be investigated to reach a general rule of thumb for when a force field is good enough, and what values of $\phi_{eff}$ enable accurate reweighting.

Another important point relates to the different ways one may improve agreement with experiments. Indeed, here we have both modified the force field (by scaling protein-water interactions) and reweighted the ensemble sampled with a given force field. Both approaches can be formulated from a statistical point of view based on Bayesian statistics [5,55], but differ in whether they have the potential of being transferable to other systems. We note, however, that when we reweight ensembles after having tuned the protein water interactions we are, in some sense, using the data twice. As we have recently noted, further studies are needed to examine the implications of this, and whether approaches can be developed that do so in a single framework [5].

## Information gain from SANS data on flexible proteins

We analysed the impact of the SANS data on the final distributions of TIA-1, and found that the SANS data only had little effect on the final distributions for the parameters $R_g$ and $D_{13}$ that we focused on. We note that these conclusions might differ for other proteins or data, or indeed for other specific questions on TIA-1. If e.g. the single domain structure was the question of interest, then contrast highlighting single domains, such at the SANS 42% contrast, provides valuable information that is orthogonal to the SAXS data. We suggested some SANS contrast situations that might provide more information gain from the SANS data, i.e. more information about the flexibility of TIA-1. Our calculations illustrated that if the differences between alternative ensembles were subtle or indistinguishable in SAXS, then they will typically also be rather small in SANS, even at optimal contrast conditions. Therefore, the obtained SANS data should be of comparable quality with the SAXS data. This can best be obtained if the incoherent scattering is reduced to a minimum, i.e. with $D_2O$ based buffer, though care should be taken to test whether the conformational ensemble is sufficiently similar in $H_2O$ and $D_2O$. In that case, it might be possible to obtain additional information, so distributions that could not be discriminated by SAXS alone could be discriminated by a combining of SAXS and SANS. Our conclusion are thus in line with previous work [47] on phospholipid nano-discs, that showed that the amount of information gained from measuring SANS data, given a model refined with SAXS, depends on parameters/questions of interest. In the present work we confirm and extend this by showing that there is additional information in the investigated SANS data, but the additional information specifically about the overall structure of TIA-1, in terms of the distribution for $R_g$ and the inter-domain distances, is limited. Nevertheless, we highlight that the improvement in agreement with the SAXS data generally mirrors improvement in the SANS data (Fig 3), suggesting that the SANS data may be used to cross-validate the SAXS-based refinement [7]. NMR paramagnetic relaxation enhancement might provide an alternative method for cross-validating transient domain-domain interactions [56].

Looking ahead, when aiming to refine an ensemble a good practical process would be first to do the simulations, and then to collect the SAXS data. The reweighting process with SAXS data can then be performed immediately after the SAXS data has been collected. If further discriminative power is needed, which of course depends on the question in mind, the more challenging SANS experiment can be designed. In that way, it is known how good the signal-to-noise ratio should be in the SANS experiment, and also at what $q$-values the most marked differences appear, such that relevant SANS setting can be chosen.

In their original study, Sonntag et al. [24] showed that when combining SAXS and SANS data they could determine more precise structural models of TIA-1, including in the presence of RNA. Here we have built upon that work, focusing only on the free TIA-1, by examining

the conformational heterogeneity of TIA-1 in solution. We have examined what information is gained from SANS, and in particular what information that comes from each of the individual SANS contrasts. There are some important differences in the our modelling approach and in that of Sonntag et al. [24], which are worth highlighting and may well affect the conclusions. Firstly, we did not include data with RRM23 deuterated and RRM1 hydrogenated [24], as there was an upturn in the data at low $q$, which may be due to slight aggregation, or the neutron beam reflecting on the sample surface. This was handled by truncation of data by Sonntag et al. [24]. Another important difference is that Sonntag et al. [24] searched for single structures to represent all data at all contrasts, i.e. each structure in their ensemble should fit all data, whereas we searched for an ensemble that fitted data when integrated. In our approach the total scattering from the reweighted ensemble fits data, whereas the scattering from individual structures in the final ensemble do generally not fit the data. Such an ensemble view makes it possible to investigate highly entropic systems where large structural variety is expected [5,6,57], but requires special care to avoid overfitting. Here, we use the BME approach for this purpose in which we balance information from the experiments with prior information encoded in the Martini energy function.

## Conclusion

We found that the latest Martini coarse-grained force field (version 3.0.beta.4.17) resulted in structures of the flexible TIA-1 that, as judged by comparison with high-quality SAXS data, were on average too compact. However, by increasing the protein-water interaction strength of the force field by about 6%, we achieved a very good agreement with the SAXS data. Reweighting the data with a Bayesian maximum entropy method further improved the fit.

In general, it cannot be expected that good agreement with data can be obtained by tuning a single parameter in a force field. Therefore, we also investigated "suboptimal versions" of the force fields, with 4% to 10% increase of the protein-water interaction strength. We stress that the term "suboptimal" here and elsewhere refers to the description of the SAXS and SANS data on TIA-1, and not the more complex problem of optimizing a transferable force field. We compared the reweighted distributions of the radius of gyration, $R_g$, and the distance between domains RRM1 and RRM3, $D_{13}$ with the reweighted distribution obtained from the "optimal" force field (with 6% increase of the protein water interaction strength). The reweighted distributions were very similar despite being rather different before reweighting. This illustrated that the BME reweighting method can be used also for suboptimal force fields. However, if the protein-water interaction strength was not increased at all, the reweighted simulations differed significantly from the others and the results were much less robust. In conclusion, the force field does not have to be perfect, but has to be "good enough", to obtain reliable results after reweighting. Whether a given force field is "good enough" can be assessed by the overlap between the initial and reweighted distribution for central parameters (in this case $R_g$ and $D_{13}$), where a substantial overlap is desirable. Moreover, the effective fraction of structures kept in the reweighted ensemble, $\phi_{eff}$, should not be too small. For this particular protein system, and these particular data, we found that the reweighted distributions were similar after reweighting when $\phi_{eff}$ was 36% or above. However, more systems need to be investigated to reach a general rule of thumb for when a force field is good enough.

Despite adding some additional information, we have shown that the structural information gain of including SANS data in the reweighting process was limited for this system and the available SAXS and SANS data. Inclusion of SANS data did not alter significantly the obtained distributions of the central parameters such as $R_g$ and $D_{13}$. It might be possible to increase the signal-to-noise ratio by decreasing the $H_2O$ content in the solvent, and thus gain

more structural information from SANS data. However, in line with previous work [47], we conclude that SAXS experiments should first be conducted and analysed, and then the SANS experiment should be carefully designed to fully benefit from the challenging sample preparation that is required for such SANS experiments with some domains deuterated and some hydrogenated.

## Supporting information

**S1 Fig. Test of faster back-mapping protocol.** Calculated theoretical form factor $P(q) = I(q)/I(0)$ for a representative frame after the full back-mapping protocol (black line) and a shortened back-mapping protocol (green). See Methods section for more details. Residuals show the relative difference.
(TIF)

**S2 Fig. Results from reweighting the simulation with overestimated protein-water interaction strength ($\lambda = 1.08$).** (A) Fit to SAXS data with adjusted force field before (black) and after reweighting at $\theta = 300$ (green). (B) $\chi_r^2$ vs. $\phi_{eff}$ for selection of $\theta$. (C) $R_g$ calculated from structures during the simulation (black), experimental $R_g$ from SAXS (red), and mean $R_g$ from the reweighted ensemble (green), with corresponding histograms in the right panel. (D) Calculated $D_{13}$ before and after reweighting.
(TIF)

**S3 Fig. Results from reweighting the simulation with overestimated protein-water interaction strength ($\lambda = 1.10$).** (A) Fit to SAXS data with adjusted force field before (black) and after reweighting at $\theta = 300$ (green). (B) $\chi_r^2$ vs. $\phi_{eff}$ for selection of $\theta$. (C) $R_g$ calculated from structures during the simulation (black), experimental $R_g$ from SAXS (red), and mean $R_g$ from the reweighted ensemble (green), with corresponding histograms in the right panel. (D) Calculated $D_{13}$ before and after reweighting.
(TIF)

**S4 Fig. Cross-validation with SANS data.** Results of SAXS reweighting (A, E; red) cross-validated with SANS at 0% $D_2O$ (B, F; black), SANS at 42% $D_2O$ (C, G; green), SANS at 70% $D_2O$ (D, H; blue). Simulated at (A, B, C, D) $\lambda = 1.04$, and (E, F, G, H) $\lambda = 1.08$.
(TIF)

## Acknowledgments

The authors would like to thank Janosch Hennig and Michael Sattler for sharing the SAXS and SANS data.

## Author Contributions

**Conceptualization:** Kresten Lindorff-Larsen.

**Data curation:** Andreas Haahr Larsen.

**Formal analysis:** Andreas Haahr Larsen, Yong Wang, Lise Arleth, Kresten Lindorff-Larsen.

**Funding acquisition:** Andreas Haahr Larsen, Lise Arleth, Kresten Lindorff-Larsen.

**Investigation:** Andreas Haahr Larsen, Sandro Bottaro, Lise Arleth, Kresten Lindorff-Larsen.

**Methodology:** Andreas Haahr Larsen, Yong Wang, Sandro Bottaro, Sergei Grudinin, Kresten Lindorff-Larsen.

**Project administration:** Kresten Lindorff-Larsen.

**Resources:** Lise Arleth, Kresten Lindorff-Larsen.

**Software:** Andreas Haahr Larsen, Sandro Bottaro, Sergei Grudinin.

**Supervision:** Yong Wang, Lise Arleth, Kresten Lindorff-Larsen.

**Validation:** Andreas Haahr Larsen.

**Visualization:** Andreas Haahr Larsen.

**Writing – original draft:** Andreas Haahr Larsen.

**Writing – review & editing:** Andreas Haahr Larsen, Yong Wang, Sandro Bottaro, Sergei Grudinin, Lise Arleth, Kresten Lindorff-Larsen.

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
