## [Decision Letter · Decision Letter 0]

28 Feb 2020

Dear Dr. Lindorff-Larsen,

Thank you very much for submitting your manuscript "Combining molecular dynamics simulations with small-angle X-ray and neutron scattering data to study multi-domain proteins in solution" for consideration at PLOS Computational Biology. As with all papers reviewed by the journal, your manuscript was reviewed by members of the editorial board and by several independent reviewers. The reviewers appreciated the attention to an important topic. Based on the reviews, we are likely to accept this manuscript for publication, providing that you modify the manuscript according to the review recommendations.

Sincerely,

Peter M Kasson

Associate Editor

PLOS Computational Biology

Nir Ben-Tal

Deputy Editor

PLOS Computational Biology

[LINK]

Reviewer's Responses to Questions

**Comments to the Authors:**

Reviewer #1: Larsen et al. present an insightful application of their Bayesian method for (minimal) ensemble reweighting. The study is well designed, the conclusions are justified by the data, and the manuscript is exceptionally clearly written. I strongly appreciate the extensive discussion on the strengths and limitations of the reweighting protocol, which will guide future efforts. Plos Comput Biol is certainly a suitable journal for this work.

I suggest only a few clarifications. Further review is not needed. Congratulations to this insightful work!

1) page 4:

"Resolution effects were included in the Pepsi-SANS calculations, using the uncertainty of the measured -values, as provided by in the fourth column of the SANS data."

How exactly is this done? The Pepsi-SANS documentation does not provide much information.

2) phi_eff gives the fraction of frames that dominate the final, reweighted ensemble. I was wondering if you can translate phi_eff into a the error of the force field (in terms of free energies). Does a certain phi_eff (e.g. 1%) mean that the force field is wrong by XX kilojoule per mole? Such translation would give phi_eff a more intuitive meaning.

3) page 17: Here you use the same term "fit" with two different meanings:

"However, the fit to the SANS data at 0% D2O worsened slightly as the fit to SAXS data improved."

I would suggest to replace with first fit ("fit to the SANS") with "agreement with SANS data" or so, since you do not really "fit" but cross-validate against SANS. This would simplify the reading. Same for the phrase "For = 1.04, the fits to SANS datasets improved..."

4) page 18: "As discussed above, although the SAXS and SANS data are fully consistent..." Further up, you write that SAXS and SANS are rather mostly (and not fully) consistent. Revise?

5) page 20: "...a slightly higher value for the protein-water interaction strength best fitted data." -> fitted THE data?

Reviewer #2: The authors report on a strategy combing coarse-grained MD using the Martini force field with SAXS/SANS data to study dynamic conformations of multidomain proteins. They demonstrate their approach with a well-studied three-domain protein TIA-1, for which high-resolution domain structures, SAXS and SANS data, including SANS data from segmentally deuterated protein, are available. The authors focus on an analysis of the dynamic ensemble of the three-domain protein in the absence of RNA, while a previous study has focused on the (more) rigid structural arrangements of the domains in the presence of RNA.

The authors performed coarse-grained MD keeping the domains semi-rigid and used a simplified back-mapping calculation to obtain an atomic description of structures to calculate Rg. The SAS and MD data were combined by Bayesian maximum entropy (BME) and quality of fits were assessed by a reduced chi_square.

The authors first optimize the regularization parameter theta, which scales the relative impact of data and simulation by reweighting in the BME approach. Thereby underestimation of calculated Rg compared to experimental data for the MD ensembles can be adjusted. Then they show that the value lambda for the protein-water interactions can be optimized in the Martini force-field to achieve best agreement with the experimental SAXS and SANS data. The authors are aware that the two adjustments for fitting experiment and simulation are not independent and may thus cause problems, but propose that the best agreement was obtained using this approach.

In a second part the author assess to role and information provided additional SANS data, when employing a MD/SAXS fitting protocol. They show that SANS can be used to aid and cross-validate the optimization of theta, while SAXS and SANS contributions are very similar. They show that the impact of contrast-matched SANS can be optimized and predicted, suggesting that in the specific example contrast-matched SANS data a sample with protonation of the second domain would provide additional information.

The manuscript is a carefully executed study combining state-of-the-art molecular dynamics simulation and BME with sparse experimental data for defining conformational ensembles of multidomain proteins. The computational procedures and analyses appear technically sound, although there are some questions (see below). The work focusses on the computational approach, while conclusions, interpretation and perhaps further validation of the structural ensembles is not attempted. Overall, the manuscript is interesting and should help to improve computational treatment of flexible multidomain proteins with SAXS/SANS data.

Thus, I recommend publication after the authors address the comments given below.

Specific comments:

- For the reweighting by optimizing theta the authors state that simulation did not include the hydration shell, while this of course contributes significantly to the experimental SAXS data. This seems a gross inconsistency. Do the author imply that the reweighting protocol compensates for ignoring this somehow? Otherwise, it seems difficult to justify to optimize a parameter while ignoring hydration shell scattering. The authors should also perform the reweighting by considering the hydration shell to assess the effect of this in their approach.

- The two fitting steps optimizing theta and lambda are not independent but rather inter-dependent as in both cases an optimal agreement between simulation and experiment is scored. How does this approach avoid a circular argument in that the two parameters are merely “fudging” in a not well-defined way? The authors seem to argue that a good force field does allow reweighting, where minimal reweighting may indicate the quality of the force field. Can this be used to generalize the approach and come up with a general recipe?

- The authors should describe which regions (residue numbers) were kept semi-rigid and which regions (linkers) were considered flexible. Is this justified, i.e. can the authors exclude that the linkers may not be completely flexible and, for example, exhibit some conformational features/propensities, or transiently interact with the domains?

- How does the dynamic ensemble compare to other experimental data (if) available. Is there a way to validate the derived ensemble by experiments?

- In the introduction, the authors refer to a number of approaches to study dynamic protein systems of based on MD and/or SAXS/SANS data. Other groups have made relevant contributions, which should be listed as well, e.g. Delaforge E, et al. J Am Chem Soc 2015. PMID 26424125; Huang JR, et al. J Am Chem Soc 2014. PMID 24734879; Bertini I, et al (2010) J Am Chem Soc doi: 10.1021/ja1063923

**Have all data underlying the figures and results presented in the manuscript been provided?**

Reviewer #1: Yes

Reviewer #2: Yes

PLOS authors have the option to publish the peer review history of their article (what does this mean?). If published, this will include your full peer review and any attached files.

Reviewer #1: Yes: Jochen S Hub

Reviewer #2: No
---

## [Decision Letter · Decision Letter 1]

13 Apr 2020

Dear Dr. Lindorff-Larsen,

We are pleased to inform you that your manuscript 'Combining molecular dynamics simulations with small-angle X-ray and neutron scattering data to study multi-domain proteins in solution' has been provisionally accepted for publication in PLOS Computational Biology.

Best regards,

Peter M Kasson

Associate Editor

PLOS Computational Biology

Nir Ben-Tal

Deputy Editor

PLOS Computational Biology

Reviewer's Responses to Questions

**Comments to the Authors:**

Reviewer #2: The authors have clarified my concerns and further improved the manuscript.

I recommend publication.

**Have all data underlying the figures and results presented in the manuscript been provided?**

Reviewer #2: Yes

PLOS authors have the option to publish the peer review history of their article (what does this mean?). If published, this will include your full peer review and any attached files.

Reviewer #2: No

---

## [Editor Report · Acceptance letter]

21 Apr 2020

PCOMPBIOL-D-19-02231R1 

Combining molecular dynamics simulations with small-angle X-ray and neutron scattering data to study multi-domain proteins in solution

Dear Dr Lindorff-Larsen,

I am pleased to inform you that your manuscript has been formally accepted for publication in PLOS Computational Biology. Your manuscript is now with our production department and you will be notified of the publication date in due course.

With kind regards,

Sarah Hammond
